# Mitochondrial Dysfunction in Pancreatic Alpha and Beta Cells Associated with Type 2 Diabetes Mellitus

**DOI:** 10.3390/life10120348

**Published:** 2020-12-14

**Authors:** Vladimir Grubelnik, Jan Zmazek, Rene Markovič, Marko Gosak, Marko Marhl

**Affiliations:** 1Faculty of Electrical Engineering and Computer Science, University of Maribor, SI-2000 Maribor, Slovenia; vlado.grubelnik@um.si (V.G.); rene.markovic@um.si (R.M.); 2Faculty of Natural Sciences and Mathematics, University of Maribor, SI-2000 Maribor, Slovenia; jan.zmazek@um.si (J.Z.); marko.gosak@um.si (M.G.); 3Faculty of Medicine, University of Maribor, SI-2000 Maribor, Slovenia; 4Faculty of Education, University of Maribor, SI-2000 Maribor, Slovenia

**Keywords:** pancreatic endocrine cells, mathematical model, mitochondrial dysfunction, cellular bioenergetics, diabetes, glucagon, insulin

## Abstract

Type 2 diabetes mellitus is a complex multifactorial disease of epidemic proportions. It involves genetic and lifestyle factors that lead to dysregulations in hormone secretion and metabolic homeostasis. Accumulating evidence indicates that altered mitochondrial structure, function, and particularly bioenergetics of cells in different tissues have a central role in the pathogenesis of type 2 diabetes mellitus. In the present study, we explore how mitochondrial dysfunction impairs the coupling between metabolism and exocytosis in the pancreatic alpha and beta cells. We demonstrate that reduced mitochondrial ATP production is linked with the observed defects in insulin and glucagon secretion by utilizing computational modeling approach. Specifically, a 30–40% reduction in alpha cells’ mitochondrial function leads to a pathological shift of glucagon secretion, characterized by oversecretion at high glucose concentrations and insufficient secretion in hypoglycemia. In beta cells, the impaired mitochondrial energy metabolism is accompanied by reduced insulin secretion at all glucose levels, but the differences, compared to a normal beta cell, are the most pronounced in hyperglycemia. These findings improve our understanding of metabolic pathways and mitochondrial bioenergetics in the pathology of type 2 diabetes mellitus and might help drive the development of innovative therapies to treat various metabolic diseases.

## 1. Introduction

Mitochondrial bioenergetics is a critical element of our life. A decline in mitochondrial function is the root cause of many inborn metabolic destructions, e.g., the Leigh Syndrome [1], and mitochondrial dysfunction has been related to several age-related diseases, e.g., type 2 diabetes mellitus (T2DM) [2,3,4], neurodegenerative [5,6], cancer [7], cardiovascular [8], and others (recently reviewed in [9]). The interrelation, particularly the causality between mitochondrial dysfunction and age-related diseases is complex, a matter of permanent discussion. Multifactorial aging processes impact mitochondrial functioning, and the loss of intrinsic mitochondrial quality and dynamics affect several other processes in the body [10]. The complexity of the interrelationship between the mitochondrial dysfunction and the diseases requires intensive research, and it is promising that the accumulation of knowledge about mitochondria and its role in pathophysiology is enormous, representing a “phase shift” in the last two decades [11].

The pathogenesis of T2DM is rather complicated and not completely understood. The link between mitochondrial dysfunction and T2DM appears to be particularly relevant for understanding the pathophysiology of T2DM. Alterations in mitochondrial structure have been found in the muscles of patients with T2DM [12]. Gene expression analyses in skeletal muscles from healthy and diabetic subjects have shown that a decreased PGC1 expression can lead to metabolic disturbances characteristic for insulin resistance and T2DM [13]. Moreover, it has been reported that myotubes with mitochondrial dysfunction exhibited insulin resistance, oxidative stress, and inflammation with impaired insulin signaling activities [14]. In subjects with T2DM, the mitochondrial DNA (mtDNA) density is mainly depleted; however, in early phases of T2DM, a compensatory mechanism can appear that increases mitochondrial gene expression even though the mtDNA copy number is reduced [15]. This compensation may be responsible for the striking contrast between the severity of mtDNA depletion and the clinical manifestation of T2DM, such as the late onset and the slowly progressive nature of the disease. Assessing mitochondrial dysfunction via the mtDNA density requires adjustments for different tissues, as the content of mtDNA varies among tissues, being relatively higher in the liver than in muscle or other tissues, for example [16].

In pancreatic tissue, a strong association between mitochondrial dysfunction and the impaired secretory response of beta cells to glucose has been established [2,15,17,18,19,20]. Disturbances in insulin secretion are related to functional and molecular alterations of beta cells, rather than a decrease in beta cell mass [21]. In T2DM, both the morphology and function of beta cells is altered. Transmission electron microscopy investigations show that mitochondria in beta cells are swollen and have disordered cristae [18,19,20]. Functionally, the hyperpolarization of the mitochondria inner-membrane potential is diminished in beta cells, partially due to UCP-2 overexpression, which causes a reduced glucose-stimulated ATP/ADP ratio, and consequently an impaired insulin secretion [18,19]. Measurements, quantifying the mitochondrial dysfunction in insulin-resistant subjects, point to approximately 30–40% decrease in mitochondrial oxidative and phosphorylation activity [16,22,23]. In beta cells, mitochondrial dysfunction has also been linked, at least partially, to the lipotoxicity-mediated suppression of glucose-stimulated insulin secretion [24]. Studying the association between diabetes and mitochondrial metabolism in pancreatic beta cells has shown that hyperglycemia markedly reduces mitochondrial metabolism and ATP synthesis [3]. In addition to the intra-mitochondrial defects, Rutter et al. [25] have pointed out the impact of a defective interconnected mitochondrial network on disturbances in insulin secretion. Moreover, they have suggested that altered mitochondrial metabolism may also impair intracellular communication and coherent multicellular activity [25]. Recently, it has been shown that in the Western-diet-induced T2DM in mice, besides the beta cells, pancreatic alpha cells undergo considerable mitochondrial alterations, affecting the T2DM-related shift in glucagon secretion [26]. The growing awareness concerning T2DM-associated mitochondrial dysfunction in both alpha and beta cells might considerably contribute to a better understanding of T2DM and may point toward new therapies for treating metabolic diseases.

Mitochondrial bioenergetics is essential for the regulation of both glucagon and insulin secretion. In our previous study, we coupled glycolysis and mitochondrial metabolism with the hormone secretion in pancreatic alpha and beta cells and presented turning on/off secretory mechanism in both cells, guided by the energy-driven metabolic switch [27]. The model includes the metabolism of both glucose and free fatty acids (FFA). At low glucose concentration, the FFA oxidation (FFAO) in mitochondria is responsible for physiologically required glucagon secretion from alpha cells. At high glucose levels, glucagon secretion is primarily switched off by the glycolytic ATP production. In beta cells, during hypoglycemia, the ATP production is low. When glucose is elevated, the mitochondrial ATP production is the primary energy source required for adequate insulin secretion. This model, linking cell metabolism with the hormone secretion, provides a useful platform for studying pathophysiology in insulin and glucagon secretion.

In the present study, we extend our previous computational models and implement the models to analyze the impact of mitochondrial dysfunction on ATP production and the consequent hormonal disorders in pancreatic alpha and beta cells. To some extent, the mitochondrial dysfunction in alpha cells has been studied previously [26]. Here, we provide a more comprehensive comparative study of both insulin and glucagon secretion in dependence on the grade of mitochondrial dysfunction. The study improves our understanding of how a decline in mitochondrial bioenergetics is linked with the typical hormonal disorders in T2DM: a decreased insulin secretion, and in-/decreased glucagon secretion in hyper-/hypoglycemia, respectively.

## 2. Model

We designed computational models incorporating intracellular mechanisms from metabolic processes to hormone secretion in the pancreatic alpha and beta cells. Specifically, models for both cell types include a detailed description of metabolite and oxygen uptake, mitochondrial bioenergetic pathways, and ATP production, as well as the ATP/ADP ratio-mediated KATP-channel activity, which in turn orchestrates the hormone secretion. Particularities of individual segments and parameters of both models were adjusted and specified to match experimental data. A more detailed description of both models is given in Section 2.1. Furthermore, we incorporate in both cellular models a T2DM-associated mitochondrial dysfunction, which leads to a reduced net flux of ATP from mitochondria and impaired hormone secretion patterns. How the effect of altered mitochondrial bioenergetics is integrated into the models is described in Section 2.2.

### 2.1. Bioenergetics and ATP-Driven Switch for Hormone Secretion

Hormone secretion in alpha and beta cells depends crucially on the metabolic pathways in the cells. Unique properties of alpha and beta cells’ metabolism enable glucose-sensing mechanisms, necessary for the modulation of glucose homeostasis. Because metabolic pathways facilitate the intracellular production of high-energy molecules ATP, the ATP concentration plays an essential role in the stimulus-secretion coupling. In the herein presented model, we divide the metabolic pathways into the glycolytic part (which can produce ATP anaerobically, in the absence of oxygen) and the mitochondrial metabolism (producing ATP at the electron transport chain (ETC) in the presence of oxygen), see Figure 1. The glycolytic and mitochondrial model components are coupled by glycolysis-produced pyruvate and NADH molecules. Whereas the ATP is produced directly by glycolysis (ADP molecules are phosphorylated during the payoff phase of glycolysis), the ATP production in mitochondria requires the ETC, driven by the oxidation of NADH and FADH_2_ molecules, acting as the electron donors of the redox reaction. Oxygen, which is consumed by this process, serves as the final electron receptor. Therefore, besides the influx of critical metabolites glucose and FFA, the model also implements the fluxes of oxygen and the intermediate fluxes of the electron donors. However, to maximally simplify the computational model, only the most abundant electron donor, namely NADH, is included, whereas the flux of FADH_2_ is considered as a part of the NADH flux.

In the process of glycolysis, each 6-carbon glucose is converted into two 3-carbon pyruvate molecules. During the first reaction of glycolysis, glucose is converted to glucose-6-phosphate (G6P). The velocity of the reaction is glucose-dependent due to the expression of low-affinity, high-Km enzyme glucokinase (hexokinase IV), in both alpha and beta cells. Consequently, the established glycolytic flux is glucose-dependent, which is crucial for the downstream energy-sensing mechanisms. Overall, glycolysis yields two molecules of anaerobically-produced ATP per one glucose molecule entering the glycolytic pathway. For the quantitative description of the glycolytic pathway, we use the theoretical framework described in Ref. [27].

On the other hand, aerobic ATP production is driven by the electrochemical proton gradient between intermembrane space and mitochondrial matrix. The proton gradient is established by the ETC, a series of complexes that transfer electrons from electron donors (reducing equivalents) to electron receptors, terminating with molecular oxygen being the final electron receptor. In the model, the reducing equivalents are the products of glucose oxidation and FFAO pathways. During glycolysis, two NAD^+^ molecules are reduced to NADH in addition to the direct (anaerobic) phosphorylation of two ADP molecules to ATP. The NADH shuttle systems transport the glycolysis-derived NADH molecules through the mitochondrial membrane, where they enter the redox reactions of the ETC (see Figure 2). The majority of reducing equivalents, originating from the glucose oxidation pathway is, however, produced by the oxidation of pyruvate, the end-product of the glycolytic pathway. The flux of pyruvate passes the mitochondrial membrane and is converted to acetyl-CoA, which enters the tricarboxylic acid cycle. Pyruvate dehydrogenase (PDH) and tricarboxylic acid cycle reactions yield 4 molecules of NADH, 1 FADH_2_, and 1 GTP, which is energetically equal to 5 molecules of NADH (for an extended explanation, see [27,28]). Following the oxidation, reducing equivalents enter the ETC, enabling aerobic ATP production.

The second metabolic pathway involved in the aerobic ATP production is the FFAO pathway. Rather than the rate of FFA entry into the FFAO reactions, the input to the model is the net rate of oxygen consumption, for which the experimental data are more widely available. In the model approximation, the net rate of oxygen consumption is fully utilized by the ETC for the oxidation of glucose- and FFA-derived reducing equivalents. Consequently, the net oxygen consumption rate can be split into oxygen consumption due to glucose oxidation and oxygen consumption due to FFAO. The net aerobic ATP production due to FFAO is proportional to the oxygen consumption due to FFAO, which is calculated as the difference between the net oxygen consumption and oxygen consumption due to glucose oxidation. For the quantitative description of the metabolic pathways in mitochondria, we utilized our previous computational models for the alpha cell [26,27] and the beta cells [27].

The rates of anaerobic and aerobic ATP production are denoted in Figure 2 by *J*_ATP,anaerobic_ and *J*_ATP,aerobic_. Glycolysis and mitochondrial respiration modulate the intracellular ATP concentration by affecting the K_ATP_ channels. Since the density of K_ATP_ channels is comparable in alpha and beta cells [29], the ATP-dependent activity of K_ATP_ channels is also similar. The increased ATP concentration during hyperglycemia lowers the K_ATP_ channel conductance, depolarizing cell’s membrane potential. In the beta cell, depolarization causes increased electrical activity, which results in higher calcium influx through voltage-dependent calcium channels, VDCC, [30] and higher insulin secretion. On the other hand, depolarization in alpha cells is followed by higher action potential (AP) frequency but lower amplitude, resulting in lower calcium influx [31] and glucagon secretion.

Our description of the alpha and beta cell’s electrical activity is based on previous computational endeavors. Specifically, for the coupling of alpha cell’s K_ATP_-channel conductance with glucagon secretion, we used model proposed by Grubelnik et al. [26], with the same parameter values. The model of glucagon granule exocytosis is founded on the theoretical framework proposed by Montefusco et al. [32]. For the beta cell, we coupled K_ATP_-channel conductance and insulin secretion as described in the model by Grubelnik et al. [27], which relies on the fundamental work of Pedersen et al. [33].

### 2.2. Mitochondrial Dysfunction and Pathophysiological Hormone Secretion

The separate modeling of aerobic and anaerobic pathways producing ATP in alpha and beta cells allows studying the effect of mitochondrial dysfunction on glucagon and insulin secretion. In cells with dysfunctional mitochondria, the net rate of aerobic ATP production is reduced, causing an insufficient lowering of K_ATP_-channel conductance. As a result, the calcium influx through VDCC during hyperglycemia is inadequately increased in the beta cell and decreased in the alpha cell [29,30,31]. Dysregulation of ATP concentration due to the mitochondrial dysfunction severely impacts hormone granule exocytosis [34,35]. The qualitative effects of the impaired mitochondrial bioenergetics are schematically presented in Figure 2. In the alpha cell, the glucagon secretion typically peaks at low glucose concentrations. However, due to impaired mitochondrial function, the peak is shifted towards higher glucose concentrations. Conversely, in beta cells, the insulin secretion rates monotonically increase with increasing glucose concentrations and the anomalous mitochondrial functioning causes the reduction of insulin secretion across the whole glucose concentration interval. The mechanisms behind the dysregulated hormone secretion in both cell types are modeled as described in more detail in the continuation.

The intracellular ATP concentration is in the steady-state approximation governed by the ATP production and ATP hydrolysis rates. The reduced net flux of ATP from mitochondria is modeled by multiplying the mitochondrial ATP flux *J*_ATP,aerobic_, as described by Grubelnik et al. [27], by an additional coefficient of mitochondrial dysfunction, *k*_md_:(1)$$J_{\mathrm{ATP}}=J_{\mathrm{ATP},\mathrm{anaerobic}}+\left(1-k_{\mathrm{md}}\right)J_{\mathrm{ATP},\mathrm{aerobic}},$$
where *k*_md_ is defined as the degree of mitochondrial dysfunction. *k*_md_ lies in the range between 0 and 1, where 0 represents a normal mitochondrial function, and 1 signifies a complete mitochondrial breakdown. According to the model by Grubelnik et al. [27], the rate of ATP hydrolysis, *J*_ATPase_, obeys the Michaelis–Menten kinetic, which is modeled by:(2)$$J_{\mathrm{ATPase}}=J_{\max,\mathrm{ATPase}}\frac{\left[\mathrm{ATP}\right]}{K_{m,\mathrm{ATPase}}+\left[\mathrm{ATP}\right]}\left(1-k_{\mathrm{ATPase},r}k_{\mathrm{md}}\right),$$
where [ATP] is the ATP concentration, *J*_max,ATPase_ is the maximal reaction rate (132 µM/s for the alpha cell, and 178 µM/s for the beta cell), *K*_m,ATPase_ is the Michaelis-Menten constant (*K*_m,ATPase_ = 2000 µM), and $k_{\mathrm{ATPase},r}$ is a scaling constant for the mitochondrial-dysfunction-dependent decrease in ATP hydrolysis. The *k*_ATPase,r_ takes on a value of 0.6. Considering Equations (1) and (2), the ATP concentration is in the steady state given by:(3)$$\left[\mathrm{ATP}\right]=\frac{K_{m,\mathrm{ATPase}}J_{\mathrm{ATP}}}{J_{\max,\mathrm{ATPase}}\left(1-k_{\mathrm{ATPase},r}k_{\mathrm{md}}\right)-J_{\mathrm{ATP}}},$$

The model Equations (1)–(3) enable us to quantify the impact of mitochondrial dysfunction on glucagon and insulin secretion in pancreatic alpha and beta cells, respectively, as we describe in more detail in the next section.

## 3. Results

Herein we present how our computational model (Appendix A) behaves under normal physiological conditions and how its output is affected by the mitochondrial dysfunction. We first focus on the mitochondrial ATP production and the so-called energy-driven metabolic switch, the main determinants for glucagon and insulin secretion in alpha and beta cells, respectively. Then, we further explore how pathologically altered ATP generation affects the hormonal secretion process.

### 3.1. Mitochondrial Function Affects ATP Production

In models for the alpha as well as for the beta cell, the ATP production depends on glycolysis and mitochondrial metabolism and is driven by glucose and FFA oxidation. Altered mitochondrial metabolism has therefore a decisive effect on the drop in intracellular ATP concentration and consequently on the secretion of the two pancreatic hormones. In Figure 3, we show how mitochondrial ATP synthesis in both cell types depends on glucose concentration. In both panels the blue lines correspond to the cell’s normal physiological function without mitochondrial dysfunction. A comparison of ATP production with regard to the cell type reveals that in both alpha and beta cells the ATP production monotonically increases with increasing glucose concentrations. However, the differences provoked by different stimulation levels are much more pronounced in the beta cell. Namely, in alpha cells at low glucose concentration, the ATP production is high compared to beta cells due to the oxidation of FFAs in mitochondria. As the concentration of glucose increases, part of FFAO in mitochondria is replaced by glucose oxidation (Figure 3a). At the same time, glucose is also metabolized by glycolysis in the cytoplasm, which is the key element that rises the ATP concentration at higher glucose levels in the alpha cell [27]. In contrast, in the beta cell the ATP production is low during hypoglycemia (Figure 3b), whereas in hyperglycemia it reaches approximately the same value as it is observed in the alpha cell. Because the production of ATP in beta cells is predominantly mediated by the mitochondrial glucose oxidation [2,36,37], this allows for very significant increase in ATP concentration when glucose levels are increased.

Results in Figure 3 also demonstrate the change in cytosolic ATP concentration due to a defective mitochondrial metabolism associated with the pathogenesis of T2DM. It is known that the mitochondrial function is decreased by 30–40% in T2DM [16,22,23]. The ATP concentration due to mitochondrial dysfunction, which is determined in Equation (1), specifically by the *k*_md_ parameter, is shown by black lines, with each line illustrating an additional decrease in mitochondrial activity by 10%. The red line illustrates a maximum of 40% mitochondrial dysfunction. It can be observed that mitochondrial dysfunction results in a drop in ATP concentration in both alpha and beta cells. In alpha cells, the decline is approximately the same for all glucose concentrations and reflects a direct decrease in mitochondrial ATP production, which is weakly modulated by glucose. This drop is due to a direct decrease in mitochondrial ATP production, which is approximately the same at low and high glucose. Namely, mitochondrial ATP production in alpha cells is close to saturated even at low glucose levels due to FFAO. The increase in cytosolic ATP concentration with increasing glucose is due to anaerobic glycolytic ATP production, which is not affected by the mitochondrial dysfunction. In contrast to alpha cells, in beta cells, a decrease in ATP production is pronounced mainly in the hyperglycemic state, which occurs primarily at the expense of increased glucose oxidation in mitochondria, which is perturbed by mitochondrial dysfunction.

### 3.2. Impaired Glucagon and Insulin Secretion in the Pathogenesis of T2DM

In the previous section we explored how the glucose concentration-dependent ATP production is affected by the mitochondrial dysfunction. Here we further investigate how T2DM-associated mitochondrial perturbations impair the secretion of glucagon and insulin secretion in the pancreatic alpha (Figure 4a) and beta cells (Figure 4b).

Under normal physiological circumstances, an increase in ATP concentration at high glucose concentration in alpha cells leads to a decrease in the conductivity of K_ATP_ channels and to cell depolarization [38]. As a result, the Ca^2+^ inflow as well as the glucagon secretion rate is lower (Figure 4a, blue line). Impaired mitochondrial ATP synthesis lowers the intracellular ATP concentration and the ATP concentrations required for glucagon secretion are reached at higher glucose values. It turns out that the mitochondrial dysfunction causes an increased glucagon secretion in hyperglycemia, whereas glucagon secretion decreases in the hypoglycemic region, as the ATP production is not sufficient to fulfil the conditions for elevated glucagon secretion. Apparently, the T2DM-associated pathological state with mitochondrial dysfunction shifts glucagon secretion to higher glucose concentrations (Figure 4a). A detailed explanation of the cellular processes underlying the switch in secretion pattern can be found in our previous work [26]. It should be noted that here, in contrast to the previously published model [26], the glycolysis is modeled on the basis of the stationary state of ATP concentration (see Model). Despite this difference in the models, both lead to qualitatively the same results.

In beta cells under normal physiological conditions, higher glucose concentrations and the corresponding increase in ATP concentration enhances electrical activity. This in turn leads to a higher calcium influx, and finally to higher insulin secretion (Figure 4b). An impaired mitochondrial ATP production lowers the insulin secretion rates, which corresponds to the decrease in the intracellular ATP level. The differences in secretion profiles are mostly pronounced in hyperglycemic conditions. Abnormalities in mitochondrial function thus lead to lower glucose-stimulated insulin secretion in beta cells, which is a hallmark of T2DM.

## 4. Discussion

Using computational modeling approaches, we analyzed how impaired mitochondrial bioenergetics affects the hormone secretion patterns in pancreatic alpha and beta cells. The model predictions show that a reduced net flux of ATP from mitochondria can importantly contribute to dysregulations in the K_ATP_-channel conductance, calcium influx through VDCC, and consequently the pathology in glucagon and insulin secretion. We have shown that mitochondrial-dysfunction-induced anomalies of hormone secretion in alpha and beta cells can be explained on the basis of very similar ATP-driven switching mechanisms. Even though the dysregulations of both glucagon and insulin secretion may result from the same pathological changes in mitochondria, they are reflected in a completely different matter in terms of glucose-dependency. Furthermore, we have shown that mitochondria play a key role in ensuring adequate ATP levels during the process of regulating hormone secretion, although their role in terms of ATP-driven switch in alpha and beta cells is different. In alpha cells, mitochondrial FFAO provides high levels of ATP, which is not significantly increased at elevated glucose levels. The modest rise in ATP concentration coincides with published measurements showing an approximately 15% increase in ATP in the hyperglycemic range [39]. Mitochondrial dysfunction thereby causes a drop in ATP (Figure 3a) and dysregulation of glucagon (Figure 4a) at both low and high glucose levels. In contrast to alpha cells, in beta cells, mitochondria provide a decisive increase in ATP due to oxidation of glucose in mitochondria (Figure 3a), which in the case of mitochondrial dysfunction leads to insulin dysregulation, especially in hyperglycemia (Figure 4b). In alpha cells, mitochondrial dysfunction causes a shift in glucagon secretion towards higher glucose concentration. The physiological values of glucagon in hypoglycemia are markedly reduced, whereas unphysiologically high glucagon levels are observed in hyperglycemia, which is a hallmark of T2DM [38,40]. In beta cells, insulin secretion is reduced over the broad range of glucose concentrations. The model results resemble the pathology of T2DM, showing that insulin secretion is indeed markedly reduced in patients with T2DM [19,41,42,43,44]. Supported by the experimental evidence that in T2DM the mitochondrial bioenergetic capacity is reduced by 30–40% [16,22,23], the model predictions considering mitochondrial dysfunction in the same range (30–40%) also matches quantitatively well the pathophysiology of glucagon and insulin secretion.

For beta cells, the results showing that the mitochondrial ATP production plays the most critical role in glucose-mediated insulin secretion are well in agreement with previous studies [2,36,45,46]. In 2019, Haythorne et al. [3] have reported that a markedly reduction in mitochondrial glucose metabolism is closely related to the pathogenesis of T2DM in mouse. This suggests that the primary cause of the insufficient insulin secretion in T2DM is the impaired mitochondrial bioenergetics, which is in contradiction to an old paradigm that gave more prominence to the beta-cell loss [3,47,48,49].

In alpha cells, the model-predicted role of anaerobic glycolysis, representing the ATP-driven switch for glucagon secretion, agrees with previous experimental observations showing that inhibition of glycolysis suppresses glucose-induced glucagon secretion [50]. In this study by Olsen et al. [50], it has also been demonstrated that the inhibition of glycolysis does not influence basal glucagon secretion. In our model, these experimental findings correspond with the FFAO in intact mitochondria, providing sufficient levels of ATP for keeping the K_ATP_ channels nearly closed and providing the glucagon secretion. Therefore, normal physiological conditions are characterized by intact mitochondria that efficiently use FFA as the main fuel. Moreover, FFA are crucial in hypoglycemic conditions for effective glucagon secretion at low glucose levels [51]. At high glucose concentrations, however, the model predictions show that glucose metabolism, particularly the anaerobic glycolytic ATP production, is the main contributor to the energy-driven switch turning off the glucagon secretion at the threshold level, which have been shown previously with a similar model [27].

It should be noted that amino acids, besides glucose and FFA studied here, also play a role in insulin and glucagon secretion. In alpha cells, an elevation in circulating amino acids causes glucagon secretion [52,53], and glucagon regulates amino acid metabolism at a systemic level [54,55]. It becomes even more evident that the liver–alpha-cell axis plays a crucial role in this regulation [55,56]. The contribution of amino acids to the energy-production in alpha cells is less relevant. In particular, in hypoglycemia under physiological conditions, alpha cells preferentially consume FFA for ATP production, in a similar way as amino acids do not fuel ATP production in hepatocytes, but instead the hepatic FFA oxidation is enhanced to supply the energy required to sustain gluconeogenesis [57]. Furthermore, amino acids were also found to affect beta cell function and insulin secretion. Amino acids such as glutamine, alanine, arginine, and others are known to cause increments in insulin secretion via activating mitochondrial metabolism similar to the downstream glucose metabolism in beta cells [58]. Therefore, the amino-acid-stimulation of insulin secretion could be in essence mathematically modeled in the same way as presented here for glucose stimulation. However, the physiological mechanisms by which amino acids confer their regulatory effects are multifactorial and involve mitochondrial metabolism [58,59] as well as other pathways such as electrical stimulation by electrogenic transport into the cells [60]. Therefore, amino acids add another layer of complexity to the islet cell metabolism and hormone secretion dynamics, and they represent a viable route for further investigation, including computational modeling approaches. Understanding the mechanisms by which amino acids along with other nutrients regulate islet hormone secretion has the potential to assess novel targets for future diabetes therapies [61].

Several studies have investigated the role of signaling pathways in alpha and beta cells, particularly the cAMP-PKA signaling pathway [62,63,64,65,66], the AMPK-signal-driven K_ATP_-channel-density regulation [67], and several other signaling pathways. For alpha cells, many of them were recently reviewed by Onyango [68]. In contrast to beta cells, where the intracellular calcium concentration is undoubtedly the main factor which drives insulin secretion, the role of calcium in alpha cells is less well defined [69]. Furthermore, it has been recently shown that fixing cAMP at high levels prevents the glucose-induced inhibition of glucagon secretion [66]. Therefore, compared to the beta cell, the intrinsic glucose-induced pathway of hormone secretion is probably more complex in alpha cells, involving a calcium-independent inhibitory pathway, paracrine signaling, and juxtracrine factors [69]. In further studies, these aspects should also be included. In this vein, mathematical modeling represents a promising approach to investigate how these additional pathways contribute to cellular function and to bioenergetic processes studied here.

The computational models are a valuable repertoire for improving of our understanding of how cellular bioenergetics impacts hormone secretion in pancreatic alpha and beta cells. In addition to the qualitative holistic understanding of the mechanisms, the model also provides quantitative insights into the relative impacts of particular sub-processes. The model predictions point to the crucial role of mitochondria, revealing how specific pathways in relation to impaired mitochondrial bioenergetics lead to the T2DM-related dysregulations of glucagon and insulin secretion. If the reduced insulin secretion in patients with T2DM is due to impaired mitochondrial bioenergetics, it would be promising to reanimate or even replace the pathologically exhaustive mitochondria. Very recently, Rackham et al. [70] reported for the first time that insulin secretion can be restored by mitochondria transfer into insulin-secreting beta cells of diabetic mouse and human islets. The mitochondria were transferred by mesenchymal stromal cells. In the experiments, it has been observed that the mesenchymal stromal cells-derived mitochondria have reached approx. 30% of beta cells, which was sufficient to reinduce a functional phenotype in the intact islets [70]. This might indicate a new epoch in clinical treatments of T2DM. For the first time, it has been shown that exactly the transfer of functional mitochondria is an important mechanism underlying the beneficial effects of the mesenchymal-stromal-cell therapies. This might even have a broader impact on a better understanding of mitochondrial bioenergetics’ role in other diseases. These therapies, and the corresponding mitochondrial transfer, are already recognized in treating injuries in the lung [71] and other tissues [72], vascular diseases [73], neurodegenerative diseases [74,75], and also COVID-19 [76,77].

Moreover, in addition to the intracellular processes that govern the function of alpha and beta cells, the pancreatic islets of Langerhans are multicellular micro-organs. Endocrine islet cells interact with each other, and this is another key player in hormonal rhythm control [69,78,79,80,81,82,83]. Beta cells are electrically coupled through gap-junctions, ensuring that the otherwise heterogeneous cell population works in synchrony [84,85,86]. In contrast, alpha cell populations’ activity is not synchronized, which collaborates the idea that these cells function individually within the islet to secrete glucagon rather than act as a syncytium [46]. However, it is known that secretion of glucagon is also influenced by local paracrine signals, mediated by beta and delta cells [69,87,88,89].

From the viewpoint of the results presented herein, the question arises of how mitochondrial dysfunction manifests itself not only at the level of individual endocrine cells but also in their socio-cellular context. It has been postulated that genetic or environmental factors, such as glucotoxicity, lipotoxicity, or inflammation, impair the beta-cell network dynamics and, hence, insulin secretion via a defective mitochondrial function [25]. A multifaceted heterogeneity characterizes beta cells, and they form subgroups with distinct metabolic properties [90,91,92]. For addressing the spatio-temporal complexity of the beta cell populations, network analyzes have been applied, which have revealed a high level of heterogeneity in beta cell connectivity [93,94,95]. Specifically, a subset of specialized and well-connected cells were identified. These so-called hub cells are not only believed to substantially affect the collective cellular activity, but were also found to have the highest rate of energy consumption [96], they are metabolically highly active, exhibit hyperpolarized mitochondria [94,97]. These highly connected cells might be strongly affected by aging [98] or T2DM pathogenesis, when the secretory demand in beta cells increases [99]. We might speculate that increasing energetic needs affect mostly the hub cells, which operate at a higher basal level of energy consumption. Moreover, it is known that realistic biological networks are in general very susceptible to hub removal [100]. This implies that impaired mitochondrial bioenergetics perturbs significantly functional beta cell networks and collective beta cell activity as well. The activity of alpha cells, under normal as well as under pathophysiological conditions, and particularly in relation to their operation in the multicellular environment, is much less explored. Some results have been provided; however, the authors emphasize tremendous heterogeneity of the alpha-cell populations [101,102]. Accordingly, in a recent computational study, it has been emphasized that considering the cell-to-cell variability is necessary when investigating how the intrinsic and paracrine factors affect the glucose-dependent glucagon release [103].

Considering both intra- and inter-cellular processes, we encourage future theoretical and experimental studies to incorporate modes of coupling among islet cells, as they represent an important consideration underlying their function. This would enable a more holistic assessment of the complex system of the pancreatic islets and the exploration of how impaired mitochondrial energetics contributes to the pathological endocrine function.

## Figures and Tables

**Figure 1 life-10-00348-f001:**
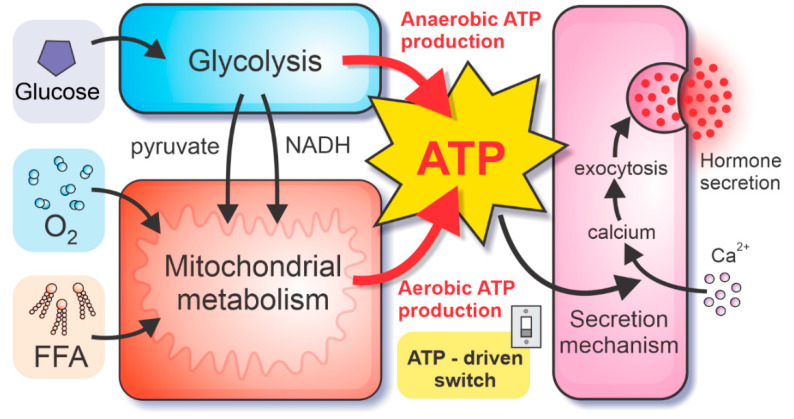
Schematic presentation of the model, coupling metabolic pathways in pancreatic alpha and beta cells with the hormone secretion. Intracellular ATP concentration drives the ATP-driven switch of hormone secretion, modulating the calcium influx and consequent calcium-dependent exocytosis of hormone granules. The ATP concentration is governed by the ATP production rates. The anaerobic ATP is yielded by direct phosphorylation of ADP during glycolysis. Conversely, the production of reducing equivalents from glycolysis, pyruvate oxidation, and FFAO drives the ETC (consuming molecular oxygen O_2_), resulting in aerobic ATP production.

**Figure 2 life-10-00348-f002:**
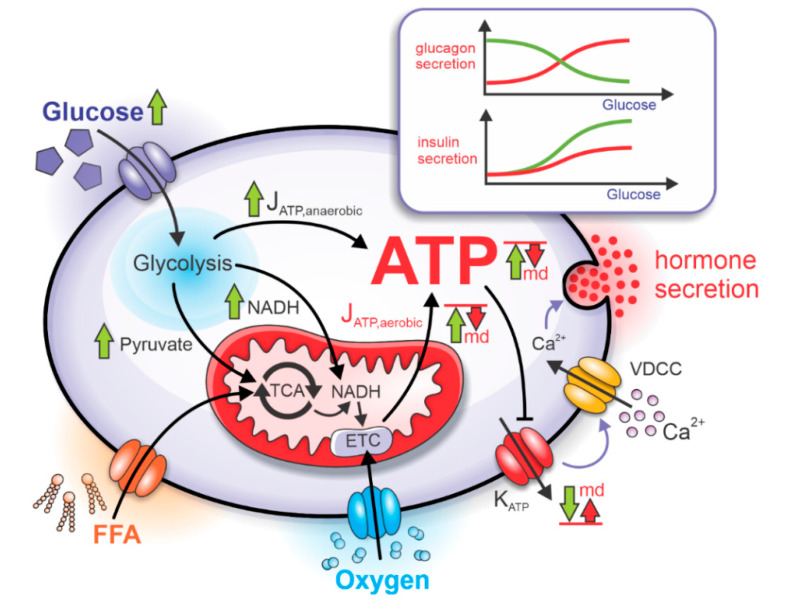
Impaired mitochondrial bioenergetics causes dysregulations in insulin and glucagon secretion. The increased glucose uptake during high plasma glucose concentration causes the acceleration of glycolytic and mitochondrial metabolism. Due to mitochondrial dysfunction (md), the aerobic ATP production rate (*J*_ATP,aerobic_) and subsequent absolute ATP concentration are reduced, causing insufficient decrease of K_ATP_ conductance and impaired effects on the VDCC. As the glucose dependency of glucagon and insulin secretion is severely impacted. Green arrows represent a response to the rise in glucose under normal physiological conditions, and red arrows represent a lack of response due to mitochondrial dysfunction.

**Figure 3 life-10-00348-f003:**
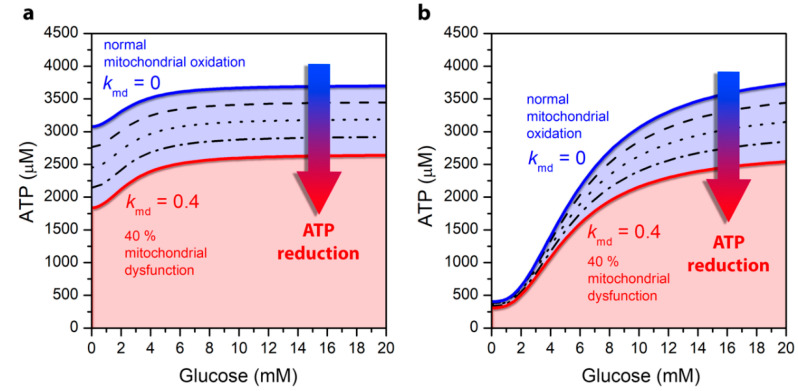
Glucose-dependent ATP concentration levels under normal mitochondrial conditions (blue line) and under different levels of mitochondrial dysfunction: 10% (dashed line), 20% (dotted line), 30% dash-dotted line, and 40% (red line), for the pancreatic alpha (**a**) and beta (**b**) cell.

**Figure 4 life-10-00348-f004:**
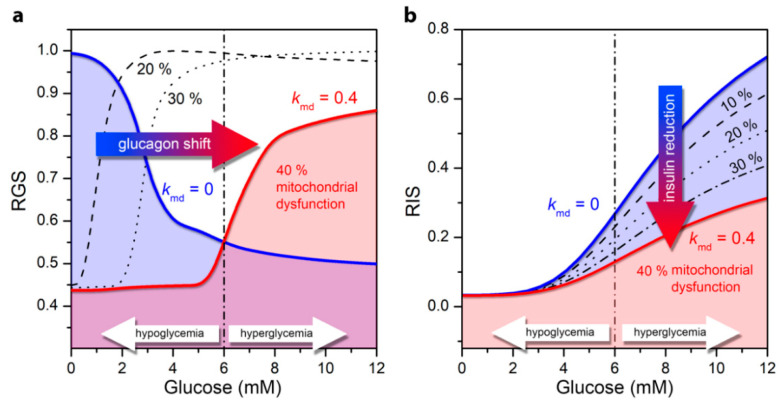
Relative glucagon and insulin secretion as a function of glucose concentration for normal cell function (blue line) and for the pathological state of T2DM under different levels of mitochondrial dysfunction: 10–40%. (**a**) Relative secretion of glucagon (RGS) in the alpha cell; (**b**) Relative insulin secretion (RIS) in the beta cell.

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
