# Peer review of "Mitochondrial Dysfunction in Pancreatic Alpha and Beta Cells Associated with Type 2 Diabetes Mellitus"

_life, 2020, doi:10.3390/life10120348_

Round 1

Reviewer 1 Report

The manuscript does not present novelty. It is an evolution of previous work (ref 46) but the presented data are not completely new and the manuscript is not well presented. The introduction is confusing and does not follow a logical thread. In addition, the results come from theoretical predictions, therefore experiments should be performed to include and consider, where possible, all factors that impact the system.

Author Response

Reviewer 1

The manuscript does not present novelty. It is an evolution of previous work (ref 46) but the presented data are not completely new and the manuscript is not well presented. The introduction is confusing and does not follow a logical thread. In addition, the results come from theoretical predictions, therefore experiments should be performed to include and consider, where possible, all factors that impact the system.

We thank the reviewer for evaluating our manuscript and regret that it was not perceived in a more positive manner. Nevertheless, we have done our best to address the comments and to improve the quality of our work.

The proposed manuscript is indeed a follow up study, but we do not agree that it does not provide novel findings. In our original work published recently in JTB (ref. 46 in the original version of the manuscript) we focused on the development of detailed mathematical models which link metabolic processes with mechanisms of glucagon and insulin secretion in alpha and beta cells. In the present study we have used the same core models but the study was performed in a completely different context. In particular, here we study how reduced mitochondrial ATP production affects hormone secretion and how do these mechanisms differ between the alpha and beta cell. In our manuscript we clearly indicate which aspects are novel, what modifications have been performed on the model, and which part of our work relies on the previous theoretical framework designed by us as well as by other authors. Moreover, it should be kept in mind that this is solely a computational study, which aims to elucidate further our understanding of metabolic pathways and pathological secretion dynamics in type 2 diabetes. Where possible, i.e. where data is available, the model was adjusted to match the theoretical results with the experimental data. But of course, in some points the model predictions go beyond the currently available and known experimental findings. Finally, we realized that our original Introduction was indeed too broad and not clear enough. In the revised version of our manuscript we have therefore shortened and reformulated the Introduction section with the aim to make it more accessible for the reader.  

Reviewer 2 Report

Introduction is too long. Equally long paragraphs. Many references and text on aging: 1 paragraph of no more than 6-8 lines. Regroup the information: talk about age, then COVID-19 and talk about age again.

In the last paragraph, the authors should remove the text referring to Results and Discussion).

What are the conclusions of the study?

Author Response

Reviewer 2

Introduction is too long. Equally long paragraphs. Many references and text on aging: 1 paragraph of no more than 6-8 lines. Regroup the information: talk about age, then COVID-19 and talk about age again.

In the last paragraph, the authors should remove the text referring to Results and Discussion).

What are the conclusions of the study?

We thank the reviewer for taking the time to review our work and for providing very constructive comments. We have done our best to address them all.

The Introduction in our original manuscript was indeed too broad and too long. We have removed or condensed several parts of the introductory section. The last paragraph was also substantially shortened, as suggested. Finally, we now more explicitly state what are the main conclusions of our study, particularly in the first paragraph of the Discussion section. All changes made to the manuscript are highlighted in the revised version of the manuscript.

Reviewer 3 Report

In this article entitled “Mitochondrial Dysfunction in Pancreatic Alpha and Beta Cells Associated with Type 2 Diabetes Mellitus” the authors explore how mitochondrial dysfunction impairs ATP synthesis and hormone secretion in alpha and beta cells. Overall, this is an interesting study that shed some light on the mitochondrial bioenergetics and metabolic pathways regulating glucagon and insulin secretion in the context of type 2 diabetes.

Please see my comments below, which are intended to improve the quality of the manuscript.

Major comments

a) Introduction section is extremely long

The Introduction is 2.5 pages long and full of information that is not necessary for the understanding of the manuscript. For instance, the authors start talking about "aging", spend two whole paragraphs (Page 2, lines 48-75) still discussing "aging", but, afterward, this is completely forgotten and the word "aging" is mentioned only once in the Discussion. Thus, why fill the Introduction with information that will not be discussed or used to put the work into perspective?

In addition, even though I completely understand our current moment, not everything needs to be about COVID-19; T2D is already a very important issue by itself. Lines 76 to 79 do not add to the study.

Besides, there is an incredibly high number of references cited only in this section: 46! For instance, in line 97 the authors cite seven articles to illustrate the association between mitochondrial dysfunction and the impaired secretory response of beta cells to glucose. I believe they could have used a fewer number of studies here.

In my opinion, the authors should reword the Introduction to focus only on important issues, such as T2D, mitochondria/mitochondrial dysfunction, and islets, as this is the aim of the study.

b) What about amino acids?

The model described by the authors only consider glucose and FFA as substrates for ATP synthesis and regulation of hormone secretion. However, several amino acids also contribute to ATP production when converted into intermediates of the tricarboxylic acid cycle. Moreover, amino acids may also play role in both glucagon and insulin secretion (Newsholme P et al., Diabetes 2006; Holst JJ et al., Diabetes 2017). Therefore, based on this knowledge, I would like to ask the authors to clarify/add to the manuscript: 1) why amino acids were not considered in the model, and 2) why amino acids were not mentioned in the discussion.

Minor comments

1) The authors use “Type 2 diabetes mellitus” or T2DM throughout the article, but “type 2 diabetes” appears twice in the Abstract (lines 15 and 24). Please correct it accordingly.

2) Page 2, line 97: Ref. 39 is not cited properly in the Reference list. Please correct it accordingly.

3) Page 4, line 159: "(…) into the glycolytic part (which can produce ATP aerobically, in the absence of oxygen) (…)" – I believe the authors mean "anaerobically" as correctly stated in Figure 1. If so, please change it accordingly.

4) Page 4, line 150: Please define "ETC" as this is the first time it appears in the text.

5) Page 5, line 193: "Glucose oxidation" was abbreviated as "GO", but this sentence appears only three times in the whole manuscript (all on the same page). Therefore, there is no need to abbreviate "glucose oxidation". Please change it accordingly.

6) Page 5, line 200: As "TCA" appears only two times in the whole manuscript (both in the same line 200), there is no need to use "TCA" instead of "Tricarboxylic acid". Please change it accordingly.

7) Page 5, line 205: "The second metabolic pathway, involved in the aerobic ATP production, is the FFAO pathway." – Please remove the commas before "involved" and after "production".

8) Page 5, line 205: "The second metabolic pathway, involved in the aerobic ATP production, is the FFAO pathway." – Please remove the commas before "involved" and after "production".

9) Page 5, lines 217-219: "Glycolysis and mitochondrial respiration modulate the intracellular ATP concentration, affecting the KATP channels, expressed in both alpha and beta cells of approximately the same density [48]." – First, please remove the comma after "channels"; secondly, I honestly did not understand what the authors mean by "alpha and beta cells of approximately the same density". Could you please clarify this point?

9) Page 5, line 222: If the abbreviation used is "VDCC", I guess it would be more appropriate to change "voltage-gated calcium channels" to "voltage-dependent calcium channels" to match the abbreviation.

10) Page 5, line 232: "(…) model by Grubelnik et al. JTB [46] (…)" – I believe this "JTB" is not necessary for this sentence.

11) Page 6, lines 238-239: "As a result, the calcium influx through VDCC during hyperglycemia is inadequately increased in the beta cell and decreased in the alpha cell." – Please cite at least one study supporting this statement.

12) Page 6, lines 240-241: "Dysregulation of ATP concentration due to the mitochondrial dysfunction severely impacts hormone granule exocytosis." – Please cite at least one study supporting this statement.

13) Page 6, line 256: Please indicate in the figure legend what the green and red arrows and lines mean.

14) Page 7, line 267: "The konstant kATPase,r takes on a value of 0.6." – First, please correct the word "constant"; secondly, please define the constant "kATPase,r" as this was not mentioned in the text.

15) Page 7, lines 279-292: Figure 3a was not mentioned in the text.

16) Page 7, lines 294-295: "Since the production of ATP in beta cells is predominantly mediated by the mitochondrial glucose oxidation (…)" – Is there any experimental evidence supporting this statement. If so, please cite at least one study supporting it.

17) Page 8, line 318: Please indicate in the figure legend what “G (mM)” means.

18) Page 9, line 346: "(…) stimulated insulin sensitivity in beta cells (…)" – I believe the authors mean "insulin secretion". If so, please change it accordingly.

19) Page 9, line 352: Please indicate in the figure legend what “G (mM)” means.

20) Page 9, lines 355-358: “The model predictions… glucagon and insulin secretion.” – The model does not show “dysregulations in the KATP-channel conductance and calcium influx through VDCC”, but that mitochondrial dysfunction leads to changes in ATP concentration and hormone secretion. I believe the authors should tone down this conclusion.

21) Page 9, line 360: “The physiological values of glucagon in hypoglycemia is markedly reduced (…)" – Here it should be “are” instead of “is”. Please correct it accordingly.

22) Page 10, lines 380-382: “Therefore, under normal physiological conditions… glucose levels [66]." – This sentence is confusing because it seems to be missing something; what is “crucial” is not clear. Please correct it accordingly.

23) Page 10, line 383: "(…) aerobic glycolytic ATP (…)" – I believe the authors mean "anaerobic". If so, please change it accordingly.

24) Page 10, lines 406-410: “While it is now clear that glucose… reviewed by Onyango [81]." – This whole sentence is confusing because it seems to be missing something. Please correct it accordingly.

25) Page 11, line 436: “(…) well-connected cells ware identified (…)" – Here it should be “were” instead of “ware”. Please correct it accordingly.

26) Page 11, lines 442-444: “Taking additionally into account… beta cell network as well." – This whole sentence is confusing because it seems to be missing something. Please correct it accordingly.

Author Response

Reviewer 3

In this article entitled “Mitochondrial Dysfunction in Pancreatic Alpha and Beta Cells Associated with Type 2 Diabetes Mellitus” the authors explore how mitochondrial dysfunction impairs ATP synthesis and hormone secretion in alpha and beta cells. Overall, this is an interesting study that shed some light on the mitochondrial bioenergetics and metabolic pathways regulating glucagon and insulin secretion in the context of type 2 diabetes.

Please see my comments below, which are intended to improve the quality of the manuscript.

We would like to thank the reviewer for a careful examination of our manuscript and for all the valuable comments. We sincerely think they helped us a lot to improve the quality of our work.

Major comments

  1. a) Introduction section is extremely long

The Introduction is 2.5 pages long and full of information that is not necessary for the understanding of the manuscript. For instance, the authors start talking about "aging", spend two whole paragraphs (Page 2, lines 48-75) still discussing "aging", but, afterward, this is completely forgotten and the word "aging" is mentioned only once in the Discussion. Thus, why fill the Introduction with information that will not be discussed or used to put the work into perspective?

In addition, even though I completely understand our current moment, not everything needs to be about COVID-19; T2D is already a very important issue by itself. Lines 76 to 79 do not add to the study.

Besides, there is an incredibly high number of references cited only in this section: 46! For instance, in line 97 the authors cite seven articles to illustrate the association between mitochondrial dysfunction and the impaired secretory response of beta cells to glucose. I believe they could have used a fewer number of studies here.

In my opinion, the authors should reword the Introduction to focus only on important issues, such as T2D, mitochondria/mitochondrial dysfunction, and islets, as this is the aim of the study.

We realize now that Introduction in the first version of our manuscript was too broad and overflown with too many citations. Accordingly, in the revised version we have substantially shortened the introductory section in which we now focus more explicitly on the issues that are addressed in our study.

Alternations made to the manuscript are highlighted accordingly.

  1. b) What about amino acids?

The model described by the authors only consider glucose and FFA as substrates for ATP synthesis and regulation of hormone secretion. However, several amino acids also contribute to ATP production when converted into intermediates of the tricarboxylic acid cycle. Moreover, amino acids may also play role in both glucagon and insulin secretion (Newsholme P et al., Diabetes 2006; Holst JJ et al., Diabetes 2017). Therefore, based on this knowledge, I would like to ask the authors to clarify/add to the manuscript: 1) why amino acids were not considered in the model, and 2) why amino acids were not mentioned in the discussion.

The reviewer has made an excellent point. In the discussion we have added a new paragraph devoted to the role of amino acids in stimulating hormone secretion in alpha and beta cells. We emphasize their systemic role at the liver-alpha-cell axis, and also their role in the cell metabolism and signaling. We discuss these previous findings in the context of our model. We highlight some basic aspects; however, a more complete insight into the role of amino acids in glucagon and insulin secretion would be gained if the particularities regarding amino acid metabolism would indeed be included into the model. Designing such a comprehensive computational model in the future would bring us closer to the ultimate goal to unlock the mechanisms by which (a mélange of nutrients) influence hormone secretion.

Minor comments

1) The authors use “Type 2 diabetes mellitus” or T2DM throughout the article, but “type 2 diabetes”appears twice in the Abstract (lines 15 and 24). Please correct it accordingly.

We have corrected this issue and write now »type 2 diabetes mellitus« in the abstract as well.

2) Page 2, line 97: Ref. 39 is not cited properly in the Reference list. Please correct it accordingly.

Corrected, thank you.

3) Page 4, line 159: "(…) into the glycolytic part (which can produce ATP aerobically, in the absence of oxygen) (…)" – I believe the authors mean "anaerobically" as correctly stated in Figure 1. If so, please change it accordingly.

Revised as suggested, thank you.

4) Page 4, line 150: Please define "ETC" as this is the first time it appears in the text.

Done.

5) Page 5, line 193: "Glucose oxidation" was abbreviated as "GO", but this sentence appears only three times in the whole manuscript (all on the same page). Therefore, there is no need to abbreviate "glucose oxidation". Please change it accordingly.

The reviewer is right, thank you.

6) Page 5, line 200: As "TCA" appears only two times in the whole manuscript (both in the same line 200), there is no need to use "TCA" instead of "Tricarboxylic acid". Please change it accordingly.

Revised as suggested.

7) Page 5, line 205: "The second metabolic pathway, involved in the aerobic ATP production, is the FFAO pathway." – Please remove the commas before "involved" and after "production".

Corrected.

8) Page 5, line 205: "The second metabolic pathway, involved in the aerobic ATP production, is the FFAO pathway." – Please remove the commas before "involved" and after "production".

Corrected.

9) Page 5, lines 217-219: "Glycolysis and mitochondrial respiration modulate the intracellular ATP concentration, affecting the KATP channels, expressed in both alpha and beta cells of approximately the same density [48]." – First, please remove the comma after "channels"; secondly, I honestly did not understand what the authors mean by "alpha and beta cells of approximately the same density". Could you please clarify this point?

The sentence has been reformulated.

9) Page 5, line 222: If the abbreviation used is "VDCC", I guess it would be more appropriate to change "voltage-gated calcium channels" to "voltage-dependent calcium channels" to match the abbreviation.

Of course, thank you.

10) Page 5, line 232: "(…) model by Grubelnik et al. JTB [46] (…)" – I believe this "JTB" is not necessary for this sentence.

Corrected.

11) Page 6, lines 238-239: "As a result, the calcium influx through VDCC during hyperglycemia is inadequately increased in the beta cell and decreased in the alpha cell." – Please cite at least one study supporting this statement.

A good suggestion. The following references have been added: Rorsman, Braun & Zhang, Cell Calcium(2012); Rorsman & Ashcroft, Physiol. Rev.(2018); Ramracheya et al., Diabetes(2010).  

12) Page 6, lines 240-241: "Dysregulation of ATP concentration due to the mitochondrial dysfunction severely impacts hormone granule exocytosis." – Please cite at least one study supporting this statement.

A good remark. We have added: Doliba et al., Am J Physiol Endocrin Metab(2012); Knudsen et al., Cell Metabol(2019).

13) Page 6, line 256: Please indicate in the figure legend what the green and red arrows and lines mean.

The figure legends have been supplemented as suggested.

14) Page 7, line 267: "The konstant kATPase,r takes on a value of 0.6." – First, please correct the word "constant"; secondly, please define the constant "kATPase,r" as this was not mentioned in the text.

Done.

15) Page 7, lines 279-292: Figure 3a was not mentioned in the text.

Thank you for noticing this flaw. The reference to the figure has been added.

16) Page 7, lines 294-295: "Since the production of ATP in beta cells is predominantly mediated by the mitochondrial glucose oxidation (…)" – Is there any experimental evidence supporting this statement. If so, please cite at least one study supporting it.

The following citations has been added: Kennedy & Wollheim, Diabetes Metabol(1998); Maechler & Wollheim, Nature(2001); Quesada et al., Diabetes(2006).

17) Page 8, line 318: Please indicate in the figure legend what “G (mM)” means.

We revised the figure so that the whole word “glucose (mM)” is now the label of the horizontal axis.

18) Page 9, line 346: "(…) stimulated insulin sensitivity in beta cells (…)" – I believe the authors mean "insulin secretion". If so, please change it accordingly.

The sentence has been reformulated.

19) Page 9, line 352: Please indicate in the figure legend what “G (mM)” means.

We revised the figure so that the whole word “glucose (mM)” is now the label of the horizontal axis.

20) Page 9, lines 355-358: “The model predictions… glucagon and insulin secretion.” – The model does not show “dysregulations in the KATP-channel conductance and calcium influx through VDCC”, but that mitochondrial dysfunction leads to changes in ATP concentration and hormone secretion. I believe the authors should tone down this conclusion.

We have reformulated the overreaching statement as suggested.

21) Page 9, line 360: “The physiological values of glucagon in hypoglycemia is markedly reduced (…)" – Here it should be “are” instead of “is”. Please correct it accordingly.

Corrected.

22) Page 10, lines 380-382: “Therefore, under normal physiological conditions… glucose levels [66]." – This sentence is confusing because it seems to be missing something; what is “crucial” is not clear. Please correct it accordingly.

The sentence has been revised.

23) Page 10, line 383: "(…) aerobic glycolytic ATP (…)" – I believe the authors mean "anaerobic". If so, please change it accordingly.

Changes as suggested.

24) Page 10, lines 406-410: “While it is now clear that glucose… reviewed by Onyango [81]." – This whole sentence is confusing because it seems to be missing something. Please correct it accordingly.

The sentence has been revised and the paragraph moved to line 557 where it better matches the context.

25) Page 11, line 436: “(…) well-connected cells ware identified (…)" – Here it should be “were” instead of “ware”. Please correct it accordingly.

The typo has been corrected, thank you.

26) Page 11, lines 442-444: “Taking additionally into account… beta cell network as well." – This whole sentence is confusing because it seems to be missing something. Please correct it accordingly.

The text has been reformulated.

Round 2

Reviewer 1 Report

Thank you for the response; I have no further comments. The authors have adequately provided a new version of the manuscript and I have no remaining concerns.

Reviewer 3 Report

First, I would like to say that the authors have done a commendable job in their revisions and have addressed my concerns.

The manuscript has significantly improved, especially the shorter Introduction and the more elaborate Discussion (I appreciate the inclusion of a paragraph discussing the role of amino acids).

I have found what I suppose it is a minor mistake that needs to be corrected.

- Page 10, lines 559-560: “For alpha cells, these many of them were recently reviewed by Onyango [68].” – I believe the word “these” is not necessary in this context.